

# Observation of ENSO linked changes in the tropical Atlantic cloud vertical distribution using 14 years of MODIS observations

Nils Madenach[1], Cintia Carbajal Henken[1], René Preusker[1], Odran Sourdeval[2], and Jürgen Fischer[1]

[1]Institute for Space Sciences, Freie Universität Berlin, Carl-Heinrich-Becker-Weg 6–10, 12165 Berlin, Germany
[2]Laboratoire d'Optique Atmosphérique, Université de Lille, Villeneuve d'Ascq, France

**Correspondence:** N. Madenach
(nils.madenach@wew.fu-berlin.de)

**Abstract.** 14 years (September 2002 to September 2016) of Aqua Moderate Resolution Imaging Spectroradiometer (MODIS) monthly mean cloud data is analyzed to identify possible changes of the cloud vertical distribution over the Tropical Atlantic Ocean (TAO). For the analysis multiple linear regression techniques are used.

Within the investigated period, no significant trend in the domain-averaged cloud vertical distribution was found. In terms of linear changes, two major phases (before and after November 2011) in the time-series of the TAO domain-average Cloud Top Height (CTH) and High Cloud Fraction (HCF) can be distinguished. While phase 1 is dominated by a significant linear increase, phase 2 is characterized by a strong, significant linear decrease. The observed trends were mainly caused by the El Niño Southern Oscillation (ENSO). The increase in CTH and HCF in phase 1, was attributed to the transition from El Niño (2002) to La Niña (2011) conditions. The strong decrease in phase 2, was caused by the opposite transition from a La Niña (2011) to a major El Niño event (2016).

A comparison with the large scale vertical motion $\omega$ at $500\,hPa$ obtained from ERA-Interim ECMWF Re-Analyses and the Nino3.4-Index indicates that the changes in HCF are induced by ENSO linked changes in the large scale vertical upward movements over regions with strong large scale ascent. A first comparison with the DARDAR data set, which combines CloudSat radar and CALIPSO lidar measurements, shows qualitatively good agreements for the interannual variability of the high cloud amount and its linear decrease in phase 2.

## 1 Introduction

One of the major sources of uncertainty in climate projection is the Cloud Radiative Feedback (CRF) (IPCC and Stocker, 2014; Bony et al., 2015) which describes the effect that clouds have on the radiation balance of the Earth. There are two main mechanisms driving cloud interaction with radiation namely the reflection of solar short-wave radiation (cloud albedo forcing) and the absorption and re-emission to space at cloud top temperature of terrestrial long-wave radiation (cloud greenhouse forcing). On global average the surface cooling by the cloud albedo forcing predominates the heating by the cloud greenhouse forcing. The strength of each forcing for a given cloud is a function of its Cloud Top Height (CTH) and its optical properties such as the Cloud Optical Thickness (COT). Low (optically thick) clouds have a negative CRF due to predominant cloud albedo forcing. High (optically thin) clouds mostly transmit incoming short-wave and trap outgoing long-wave radiation leading to a positive



CRF (e.g., Graham, 1999).

For this work the focus lies on the Tropical Atlantic Ocean (TAO). The tropical Atlantic ocean-atmosphere system is largely influenced by the Hadley-Walker circulation. Large deep convective systems with elevated Cirrus clouds at the convective out-flow predominate at the Intertropical convergence zone and shallow cumulus clouds at the trade wind regions. The interannual

variability of climate variables at the tropical Atlantic is coupled with the El Niño Southern Oscillation (ENSO). An overview of ENSO linked influences on Earth's climate system can be found in e.g. (Timmermann et al., 2018). In order to better understand questions as e.g. how clouds, circulation and climate interact (Bony et al., 2015), this region is of major interest of recent research efforts. At the Barbados Cloud Observatory long-term ground based measurements are done (e.g., Stevens et al., 2016). In December 2013 and August 2016 the Narval campaigns within the **H**igh **D**efinition **C**louds and **P**recipiation

for advancing **C**limate **P**rediction (HD(CP)$^2$) project (Klepp et al., 2014) produced a large amount of airborne based measurements for this region. Furthermore, the study area is one simulation domain of the cloud resolving ICON-LEM (**ICO**sahedral **N**onhydrostatic-**L**arge **E**ddy **S**imulation) model developed within the HD(CP)$^2$ project (Dipankar et al., 2015; Heinze et al., 2017).

In this work 14 years of the **MOD**erate-resolution **I**maging **S**pectroradiometer (MODIS) level 2 cloud product (Platnick et al.,

2015) were analyzed using Multiple Linear Regression Model (MLRM) techniques in order to get insights of the long-term variability and possible trends within the CTH and cloud vertical distributions due to a warming climate. The passive imager MODIS aboard the polar orbiting NASA satellites Terra and Aqua provide the possibility to obtain observational information about the cloud vertical distribution on high temporal (on climate scales) and spatial resolution. MODIS aboard Aqua delivers reliably high resolution data of cloud properties since 2002 (Platnick et al., 2015). Every point on Earth is seen every one

to two days. For the evaluation of the MODIS observations, the DARDAR (CloudSat RA**DAR** and CALIPSO LI**DAR**) data set which relies on information of active instruments were used (Delanoë and Hogan, 2010). Further more we analyzed the level 3 monthly mean Sea Surface Temperature (SST) and Total Column of Water Vapor (TCWV) from **A**dvanced **M**icrowave **S**canning **R**adiometer - **E**arth Observing System (AMSR-E) aboard Aqua (Wentz, 2004). For the interpretation of the results we examined Era-Interim reanalysis data of the vertical velocity $\omega$ at $500\,hPa$ (Dee et al., 2011).

In Sec. 2 we will give an overview of the data used for the analysis. Section 3 introduces the methodology used. In Sec. 4 the results are presented and discussed, and in Sec. 5 the work is summarized and some conclusions are drawn.

## 2   Data

In this section an overview of the used data and its processing is given. The study area is defined by a latitude-longitude box of $30°$ S–$30°$ N and $70°$ W–$20°$ E and is hereinafter referred as tropical Atlantic. The study period ranges from September 2002

to September 2016.



## 2.1 MODIS

MODIS measures reflected shortwave and emitted long-wave radiation in 36 spectral bands. MODIS is aboard the NASA
Earth Observing System (EOS) satellites Aqua (1:30 p.m. LT ascending node) and Terra (10:30 a.m. LT descending node).
Both satellites are in a polar sun-synchronous orbit at an altitude of $705\,km$. In this work solely data from MODIS aboard
Aqua (Platnick et al., 2015), which is part of the Afternoon Train (A-train) constellation, is considered. With a swath of
$2330\,km$ (cross track) by $10\,km$ (along track at nadir) and a scan rate of 20.3 rpm (cross track) global coverage is acquired
almost daily. The level 2 data of collection 6 (Platnick et al., 2015) provides cloud optical and microphysical property data at a
spatial resolution of $1\,km$ and cloud top property data at $5\,km$ as well as at $1\,km$ resolution (Menzel and Strabala, 2015; Platnick
et al., 2017). For the analysis only data from overpasses during day time (ascending node) were used. The vertically resolved
cloud fractions were calculated using the $1\,km$ cloud mask and the International Satellite Cloud Climatology Project (ISCCP)
cloud classification scheme explained in Sec. 3.1.

In order to get daily composites from the MODIS overpasses the segments were regridded on a regular $0.1° \times 0.1°$ grid. For
every grid cell the mean, minimum, maximum and the variance of the containing pixels were computed. Due to the proximity
to the equator of the analyzed region for the majority of the grid cells, the pixels arise from a single overpass. To account for
possible miss-classification of the MODIS cloud mask (clear sky conservative) due to e.g. cloud edges, broken clouds, smoke,
dust or sun-glint, pixels that were poor retrieval candidates were excluded using the clear sky restoral flag (Hubanks, 2015).
After computing the daily composites for every day of the investigated period, grid cell based monthly means were computed
and afterwards regridded to a $0.2° \times 0.2°$ grid. Based on the monthly means, climatologies were calculated by averaging every
month of the year over the 14 years of MODIS data. Monthly anomalies were produced by subtracting the climatology from
every single monthly mean.

## 2.2 DARDAR

To compare the results obtained with MODIS additionally data for Total Cloud Fraction (TCF), High Cloud Fraction (HCF),
Middle Cloud Fraction (MCF) and Low Cloud Fraction (LCF) computed from DARDAR data (Delanoë and Hogan, 2010) also
using the ISCCP cloud classification scheme were analyzed. DARDAR is a project from Laboratoire Atmosphères, Milieux,
Observations Spatiales and the Cloud Group of the Department of Meteorology at the University of Reading to retrieve cloud
properties by combining CloudSat RA**DAR** and CALIPSO LI**DAR** measurements. Both the CloudSat and CALIPSO satellite
fly within the A-train. CloudSat lags Aqua by between 30 seconds and 2 minutes and CALIPSO lags CloudSat by no more
than 15 seconds. The data is available for the period from June 2006 to June 2016 with gap from May 2011 to April 2012. Due
to the much lower sampling of the active lidar and radar instruments the monthly means were computed on a $2° \times 2°$ grid.

## 2.3 AMSR-E

Furthermore, data from the AMSR-E instrument aboard Aqua was used to acquire information about the SST and TCWV.
AMSR-E measures the brightness temperature at 6 different wavelengths in the microwave range between between 0.34 and





4.35 cm. For every wavelength, the horizontal and vertical polarized radiation is measured leading to 12 channels in total. The spatial resolution depends on the the channel and varies from $5.4\,km$ to $56\,km$ . The radiometer has a viewing swath width of $1445\,km$ and an incidence angle of $55°$. For the analysis the SST and the TCWV from version 2 of the monthly Level-3 product (AE_MoOcn) with a spatial resolution of $0.25° \times 0.25°$ were used (Wentz, 2004). The AMSR-E data is available from June 8th 2002 through October 4th 2011.

### 2.4 ERA-Interim

To get insights about the predominant large scale dynamics, the vertical velocity $\omega$ at $500\,hPa$ was acquired from the Era-Interim reanalysis (Dee et al., 2011). In order to roughly match the MODIS overflight time, the daily Era-Interim analyses data at 12:00 was used. From the daily data ($0.2° \times 0.2°$), monthly means were computed the same way as explained in Sec. 2.1.

## 3 Methods

Following, the ISCCP cloud classification scheme is introduced. Furthermore, the MLRM used for the time-series analysis is described.

### 3.1 Assessment of the cloud vertical distribution

The ISCCP cloud classification scheme is widely used to simply distinguish between different cloud types using remote sensed information. The ISCCP is a project, started 1982 as part of the World Climate Research Programme (WCRP), with the goal to collect and analyze satellite radiance measurements to infer the global distribution of clouds, the cloud properties, and the cloud diurnal, seasonal, and interannual variations. As Fig. 1 illustrates the clouds are distinguished by its Cloud Top Pressure (CTP) in $hPa$ and its optical thickness. There are three categories for the vertical distribution namely high (CTP $< 440\,hPa$ ), middle ($\leq 440$ CTP $< 680\,hPa$ ) and low (CTP $\geq 680\,hPa$ ). For the MODIS data the cloud height category flag from the MODIS $5\,km$ quality assurance (Hubanks, 2015), that is based on the explained ISCCP thresholds was used to discriminate between the height categories.

### 3.2 Multiple linear regression

To obtain information of linear changes within the data-sets a simple MLRM was developed. For the model development, the time series of the analyzed parameters ($y$) were assumed to be composed of a constant $\mu$, a seasonal $S$, a trend $\tau$ and a noise $\eta$ component. These assumptions lead to the following model equation.

$$y(t) = \mu + S(t) + \tau \cdot t + \eta(t) \tag{1}$$

Where the seasonal component was specified by the sum of sine and cosine with a period of 12 months.

$$S(t) = \sum_{i=1}^{2} \sin\left(\frac{2\pi it}{12}\right) + \cos\left(\frac{2\pi it}{12}\right) \tag{2}$$





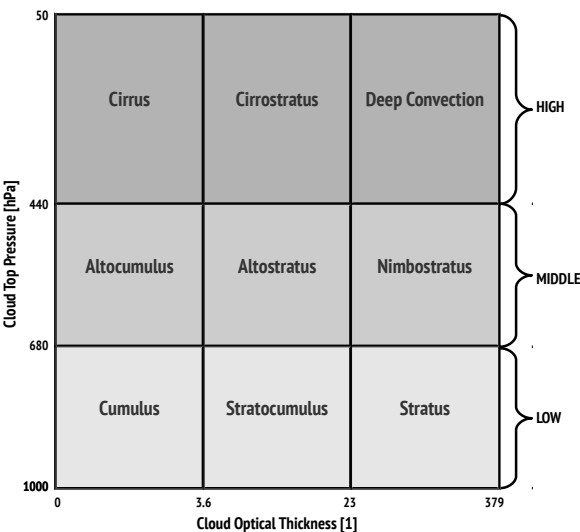

**Figure 1.** ISCCP cloud classification scheme (Rossow and Schiffer, 1999).

Due to bi-annual variations in the time series the first harmonic (i=2) of sine and cosine was added to the model.
In a general form Eq. 1 can be written as:

$$y(t) = \mathbf{x}_t^T \mathbf{b} + \eta_t \tag{3}$$

Where $\mathbf{x}_t^T$ represents a vector including all explanatory parameters required to describe the parameter $y$ at time step $t$ and $\mathbf{b}$ a
5   vector containing all regression coefficients. In Eq. 4 all possible T linear combinations of the n parameters are summarized in
matrix notation.

$$[\mathbf{y}] = [\mathbf{X}][\mathbf{b}] + [\boldsymbol{\eta}] \tag{4}$$

Here $\mathbf{y}$ stands for the $T{\times}1$ vector with the measured parameter for all $T$ month, $\mathbf{X}$ for the $T \times n$ model matrix including all
predictor variables for the estimation for all $T$ months and $\mathbf{b}$ for the $n{\times}1$ vector containing all the regression coefficients. The
10   variations that are not explained by the model are stored in the $T{\times}1$ noise vector $\boldsymbol{\eta}$.

To find the vector, which best fits to the data (denoted as $\hat{\boldsymbol{b}}$), the ordinary least square method is used. This method tries to
predict the expected parameter $\hat{y}$ by minimizing the sum of squares of the distances between measured and estimated values.
The remaining noise between prediction and measurement is termed residual:

$$[\boldsymbol{\eta}] = [\mathbf{y}] - [\hat{\boldsymbol{y}}] \tag{5}$$





It is important to remember that $\mathbf{y}$ is the vector with the measured values and $\hat{\mathbf{y}}$ is the vector containing the values estimated by the model. Consequently, the following expression has to be minimized, in order to minimize the sum of squared residuals:

$$M = ||\boldsymbol{\eta}||^2 = ||\mathbf{y} - [\hat{\boldsymbol{y}}]||^2 = [\boldsymbol{\eta}]^T[\boldsymbol{\eta}] \tag{6}$$

With $[\hat{\boldsymbol{y}}] = [\mathbf{X}] [\hat{\boldsymbol{b}}]$ and some simplifications Eq. 6 results in:

$$M = [\mathbf{y}]^T[\mathbf{y}] - 2[\hat{\boldsymbol{b}}]^T[\mathbf{X}]^T[\mathbf{y}] + [\hat{\boldsymbol{b}}]^T[\mathbf{X}]^T[\mathbf{X}][\hat{\boldsymbol{b}}] \tag{7}$$

To find the values of the vector $\hat{\boldsymbol{b}}$, including all regression coefficients, which minimizes the sum of squared residuals, the derivation of $M$ with respect to $\hat{\boldsymbol{b}}$ must be zero:

$$\frac{\partial M}{\partial \hat{\boldsymbol{b}}} = -2[\mathbf{X}]^T[\mathbf{y}] - 2[\mathbf{X}]^T[\mathbf{X}][\hat{\boldsymbol{b}}] = 0 \tag{8}$$

After transposing Eq. 8, the following linear equation system in matrix notation must be solved in order to obtain $\hat{\boldsymbol{b}}$:

$$[\mathbf{X}]^T[\mathbf{y}] = [\mathbf{X}]^T[\mathbf{X}][\hat{\boldsymbol{b}}] \tag{9}$$

Multiplying both sides with $([\mathbf{X}]^T[\mathbf{X}])^{-1}$ leads to the final equation to be solved:

$$[\hat{\boldsymbol{b}}] = ([\mathbf{X}]^T[\mathbf{X}])^{-1}[\mathbf{X}]^T[\mathbf{y}] \tag{10}$$

To get an idea of the the linear correlation between two variables $x$ and $y$ the Pearson Correlation Coefficient (PCC) was used. It is defined as the ratio of the co-variance of $x$ and $y$ and the product of the two standard deviations $s_x$ and $s_y$:

$$PCC(x,y) = \frac{cov(x,y)}{s_x s_y} \tag{11}$$

The PCC is bounded by -1 and +1. For PCC = -1 there is a perfect, negative linear association between $x$ and $y$ and for PCC = +1 there is a perfect positive linear association between the two variables. For PCC = 0 there is no linear correlation, but there might be non linear correlations between $x$ and $y$.

A more detailed explanation of the used MLRM can be found in e.g. Wilks (2011).

**4   Results**

**4.1   Monthly means and climatologies**

In Fig. 2 the monthly means for January (left) and July (right) 2009 are displayed for the CTH (upper panels) and the TCF (lower panels) for the investigated spatial domain of the tropical Atlantic. Near the equator the data reveals a band with high cloud tops and high cloud amounts indicating the ITCZ. Furthermore the seasonal shift of the ITCZ is represented. Lower

cloud tops due to the well known trade wind inversion are predominant in the trade wind regions. The stratocumulus region near the coast of Angola and Namibia with higher CF and low CTH are illustrated in the data as well. In boreal winter (left) a





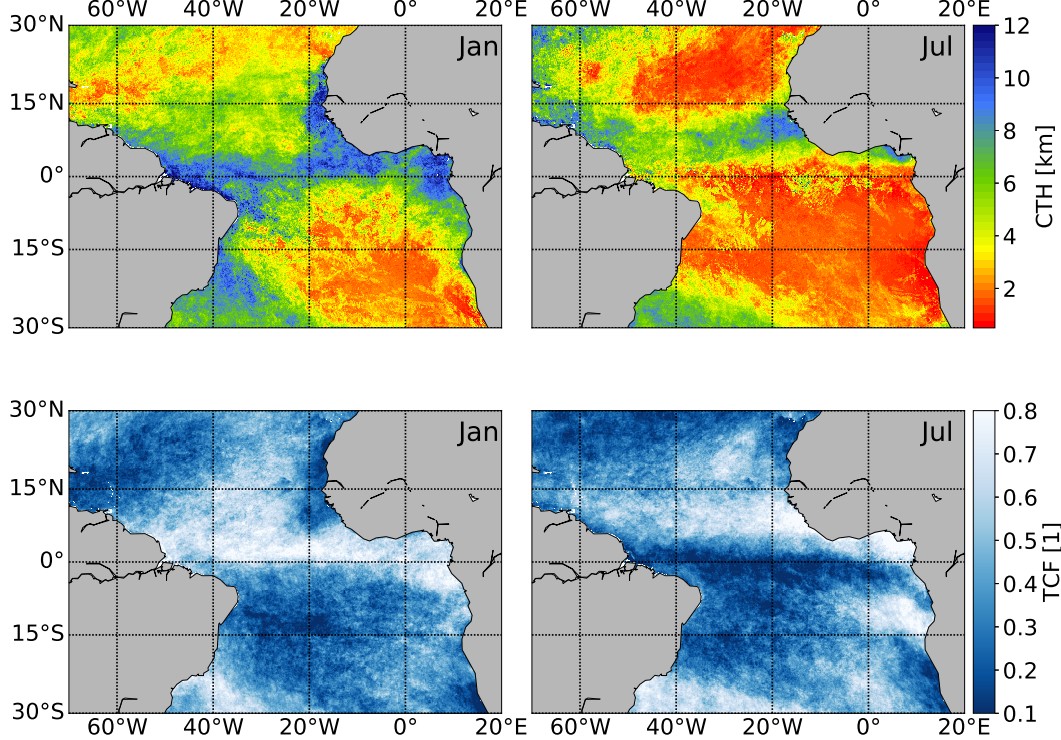

**Figure 2.** Monthly means with a resolution of $0.2° \times 0.2°$ based on $0.1° \times 0.1°$ daily composites for CTH (upper) and TCF (lower) from January (left) and July (right) 2009.

part of the south Atlantic convergence zone can be identified southeast of Brazil with high cloud tops and high cloud amounts. The tropical Atlantic domain-averaged climatologies (see Sec. 2) of the CTH and the vertically resolved CF's are shown in Fig. 3. Solid lines indicate MODIS and dashed lines DARDAR data. The CTH climatologies vary between $3.6\,km$ in boreal summer and $5.5\,km$ in boreal spring and in December respectively. The climatologies of the CF's show that the MCF is negligible low (not shown) and the TCF is roughly composed for the same portion of LCF and HCF. The HCF is in phase with CTH showing similar seasonal variability. The LCF and CTH show a negative correlation. This indicates that the mean seasonal variability of the CTH is mainly related to changes in the vertical cloud distribution. The climatologies calculated for the DARDAR data (June 2006- June 2016) reflect the same seasonal variability, indeed they have a positive bias towards higher CF's which can be explained with the higher sensibility of lider for high, optically thin clouds.

## 4.2 Anomalies

To get insights of the interannual variability and the temporal evolution of the variables, the monthly tropical Atlantic three month running mean anomalies (see Sec. 2) were analyzed. As illustrated in Fig. 4 the mean anomalies of CTH are strongly





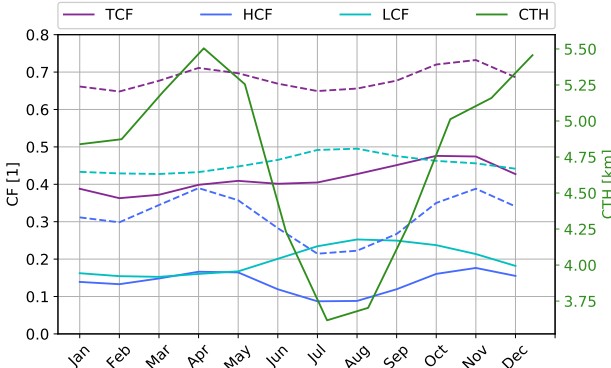

**Figure 3.** Domain-averaged TAO climatologies of TCF, HCF, LCF and CTH. The dashed lines indicate climatologies of the DARDAR-Data (2006.06-2016.06) and the solid lines MODIS-Data (2002.09-2017.06).

positive correlated with the HCF anomalies, with a PCC of 0.89 and an $R^2$-value of 0.8. The interannual variability of CTH is mainly driven by the HCF. The time-series of the CTH and the HCF show major phases with negative anomalies around 2002, 2009 and at the end of the period starting in 2015 and major phases with positive anomalies between 2007 and 2008 as well as between 2010 and 2011. Remarkable is the large anomaly of the CTH and HCF at the end of the time-series which is not

present in the TCF and LCF. The times where the anomalous phases occur indicate a link to the ENSO. An impact of ENSO to the tropical Atlantic ocean-atmosphere system is well known even though the exact mechanism of teleconnection is still not fully understood. Positive ENSO events were found to be associated with higher SSTs in the tropical Atlantic (Klein et al., 1999), a weakening of the Walker circulation and the Atlantic Hadley circulation (Klein et al., 1999; Wang, 2004, e.g.) and a stronger vertical wind shear over the tropical Atlantic (Zhu et al., 2014). The lower panel of Fig. 4 displays the three month

running mean of the HCF anomaly and additionally the Nino3.4-Index (obtained from the NOAA climate prediction center). The interannual variability of the mean HCF at the tropical Atlantic and the Nino3.4-Index are negatively linked (PCC=-0.53). Where negative ENSO events are associated with more high clouds and vice versa. As shown in (Marchand, 2013) a strong negative correlation between ENSO and high cloud amount is also present in MODIS (and **M**ulti-angle **I**maging **S**pectro**R**adiometer (MISR)) observations at the Tropical Warm Pool region (ocean between 30° N–30° S and 100° E–160° E),

the Indian Ocean as well as the Tropical Western and Central Pacific. Due to the strong ENSO influence, two phases can be distinguished within the time series, with an overall increase of the high cloud amount from September 2002 (El Niña event) to end of 2011 (La Niña event) and a decrease from end of 2011 to 2016 (strong El Niño event).

## 4.3 Multiple linear regression analysis

In order to get insights of possible linear changes in the investigated variables associated with a warming ocean and changes

in atmospheric dynamics due to climate change, a multiple linear regression analysis, explained in more detail in Sec. 3.2, was



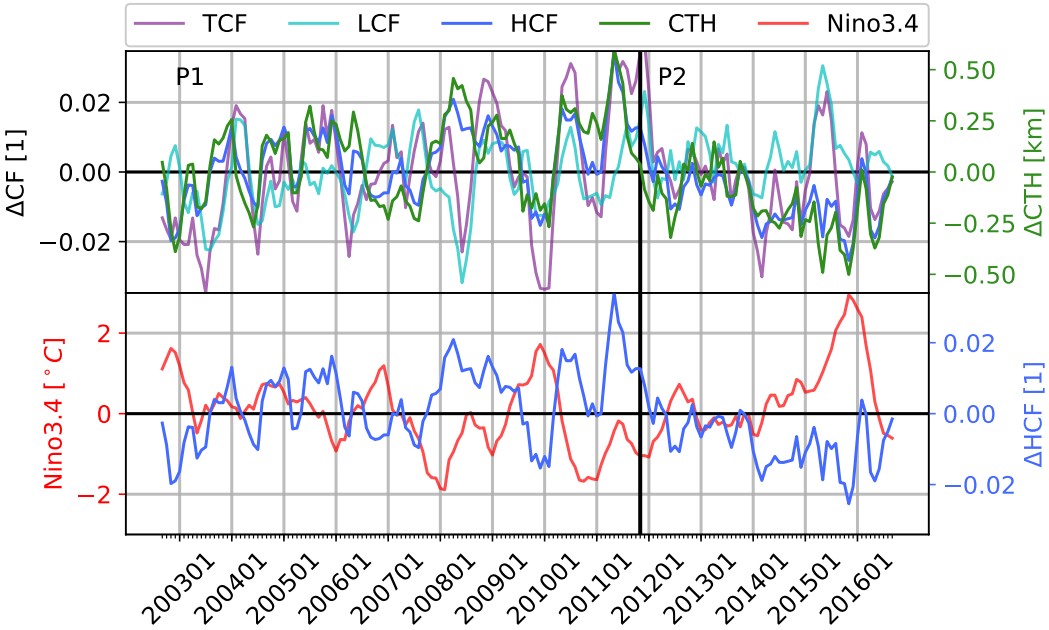

**Figure 4.** Upper panel: time-series of the tropical Atlantic three month running mean anomalies of CTH, TCF, HCF and LCF. Lower panel: as upper but comparing the HCF and the Nino 3.4-Index (NOAA climate prediction center). The PCC is found to be -0.53.

performed. For the analyzed time period no consistent significant linear trend was found for the domain-average (not shown). The time series (especially CTH and HCF) are highly influenced by the ENSO. As the time-series showed opposed linear trend at the end of the time-series compared to the beginning, it was separated into two phases and the model was applied to both phases. To find the best separation of the phases the maximum of the sum of the explained variances of the two models was

5     calculated for differing phase separations. The best fit was found for the period from September 2002 to November 2011 (P1) and the period from November 2011 to June 2016 (P2). The changes in P1 are mainly associated with a large La Niña event (2010) at the end of P1 and that in P2 with a large El Niño event (2015) at the end of P2. Figure 5 illustrates the results of the regression analysis applied on the tropical Atlantic mean for the different variables. The black line shows the data and the blue line the regression model. The slope $\tau$ is given per decade (dec) and tagged with an asterisk in case of significance (p<0.05).

10     In addition, the Nino3.4-Index is displayed in red.

The CTH in Fig. 5 (a) and the HCF in Fig. 5 (c) show a significant El Niño related CTH decrease in P2 of $579\,m\,dec^{-1}$ and of $0.024\,dec^{-1}$ for HCF, respectively. This indicates that observed changes in the CTH are mainly due to changes in high cloud amount. The LCF (d) shows neither significant changes in P1 nor in P2. The changes of the TCF in P2 are visible, but statistically not significant (b). In P1 all variables (a-d) show a linear increase which are all significant, except for the LCF.

15     To evaluate the anomalies observed with the passive MODIS instrument, DARDAR data (see Sec. 2.2) from active LIDAR







**Figure 5.** Time-series of the tropical Atlantic means. a) CTH, b) TCF, c) HCF, d) LCF, e) $\omega_{500}$. In black the MODIS data is displayed and in blue the applied MLRM. In red the Nino3.4-Index is displayed.

and RADAR instruments was analyzed. Due to the low sampling, monthly means were computed and compared to MODIS with a $2° \times 2°$ resolution. As Fig. 6 illustrates the seasonal and interannual variability of the HCF, as well as its decrease in P2,




is despite of the low sampling visible in the DARDAR data as well. As already seen in the climatologies (Fig. 3) DARDAR has a positive offset.

To get spatial information of the anomalies (trends) the MLRM was applied to every $0.2° \times 0.2°$ grid-box. Fig. 7 displays the

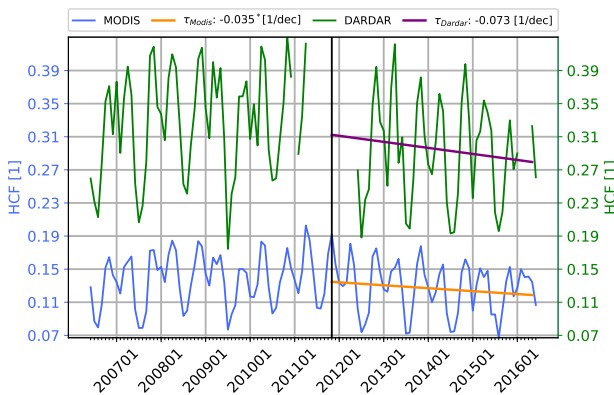

**Figure 6.** Comparison of the TAO domain-average of the $2° \times 2°$ DARDAR (green) and MODIS (blue) HCF monthly means.

linear part of the MLRM $\tau$. Black plus signs indicate that within a $1° \times 1°$ grid-box more than 50% of the 25 anomaly values
are significant. The CTH (a) and the HCF (d) show very similar patterns in both phases with significant increases between
$\pm 10°$ N/S in P1 and significant decreases between 0 and $20°$ N in P2 confirming the strong link between changes in CTH and
the HCF. For the TCF (b) a significant decrease in P2 at the stratocumulus region near the coast of Angola and Namibia, which
is caused by a decrease in LCF (c) is visible. The HCF and the LCF have opposing patterns that causes most of the signal in
TCF to be extinguished. This endorses the idea to look at the cloud vertical distribution rather than just on total cloud amount
in climate studies.

## 4.4    Link between TAO and ENSO

During a positive ENSO phase the anomalous high SST in the Pacific induces an average warming of the tropical Atlantic
troposphere, e.g. trough Kelvin waves causing an increase in the meridional tropospheric temperature gradient (e.g., Horel and
Wallace, 1981; Yulaeva and Wallace, 1994; Chiang and Sobel, 2002; Zhu et al., 2014). This leads to an increase of the vertical
wind shear (Aiyyer and Thorncroft, 2006; Shaman et al., 2009; Zhu et al., 2014, e.g.) and tends to increase the static stability
over the tropical Atlantic (e.g., Tang, 2004; Larson et al., 2012). Moreover, a weakening of the Atlantic Hadley and Walker
circulation and an east-ward shift of the latter (Klein et al., 1999; Wang, 2006) has been associated with higher Pacific SSTs.
Furthermore, a reduction in equatorial Atlantic rainfall (Saravanan and Chang, 2000; Chiang and Sobel, 2002), a lagged SST
increase caused by a warmer troposphere and reduced latent and sensible heat losses due to weaker trade winds and a reduction
in cloudiness (Curtis and Hastenrath, 1995; Enfield and Mayer, 1997; Klein et al., 1999; Saravanan and Chang, 2000; Huang,





**Figure 7.** Trends of CTH, TCF, HCF, LCF for every $0.2° \times 0.2°$ grid box. At the left the trends for phase 1 and at the right trends for phase 2 are displayed. The plus signs indicate that within a grid box of $1°$ more than half of the trends were statistically significant. The slopes are displayed per decade.



**Table 1.** Definition of the $\omega_{500}$-bins

| bin # | vertical motion | $\omega_{500}$ range [Pa s$^{-1}$] |
|---|---|---|
| bin-1 | strong subsidence | $\omega_{500} > 0.15$ |
| bin-2 | moderate subsidence | $0.05 < \omega_{500} \leq 0.15$ |
| bin-3 | no/weak vertical motion | $-0.05 < \omega_{500} \leq 0.05$ |
| bin-4 | moderate ascend | $-0.15 < \omega_{500} \leq -0.05$ |
| bin-5 | strong ascend | $\omega_{500} \leq -0.15$ |

2002) were found during El Niño. The reduction in cloudiness leads to changes in radiation fluxes due to higher absorption of solar radiation by the ocean, which in turn increases the mean tropical Atlantic SST (Curtis and Hastenrath, 1995; Klein et al., 1999) strengthening the positive SST feedback between the Pacific and the Atlantic. In general the described effects are observed to be inverted during La Niña.

As shown in Sec. 4.2 & Sec. 4.3 a link between ENSO and TAO cloudiness, primarily in high cloud amount, was observed in our analysis as well. In addition, the analysis of AMSR-E retrieved SST and TCWV data show a consistent (Klein et al., 1999, e.g.) three months delayed TAO SST and TCWV increase after an El Niño event (Fig. 8). Further more, a significant positive trend of 0.3 °C dec$^{-1}$ was found for the regional mean SST.

To investigate whether the observed anomalies in HCF are linked to ENSO induced changes in the large scale circulation over
the TAO, the large scale vertical velocity $\omega$ at 500 $hPa$ acquired from ERA-Interim data was analyzed. The results are illustrated in Fig. 5 (c) for the TAO mean and in Fig. 7 (e) for the grid-box based analysis. $\omega$ is negatively defined so that negative velocities imply upward movements and vice versa. For the TAO mean a decrease in P1 and an increase in P2 was found. The anomalies are mostly due to strengthening/weakening in upward movement rather than weakening/strengthening in downward movement. This is coherent with the results of the grid-box based analysis displayed in Fig. 7 (e) where in both phases the strongest and
most of the significant anomalies are found at the ITCZ region where upward movement predominates.

The comparison of the patterns of $\Delta$HCF (Fig. 7 (d)) and $\Delta\omega_{500}$ (Fig. 7 (e)) reveal similar patterns for both phases. The increase/decrease in the high cloud amount in P1/P2 seems to be associated with an increase/decrease in large scale upward motion in equatorial Atlantic ocean. A stratification of the HCF into different $\omega_{500}$-bins as listed in Table 1 confirms that the decrease of the high cloud amount in P2 occurs in regions with strong upward movements, Fig. 9 (d) and Fig. 9 (e). This
supports the idea that the observed anomalies, particularly the decrease of CTH/HCF in P2, are caused by an ENSO induced east-ward shift of the Walker circulation and a weakening of the Walker and the Atlantic Hadley Cell and indicate an increased vertical stability and vertical wind shear.





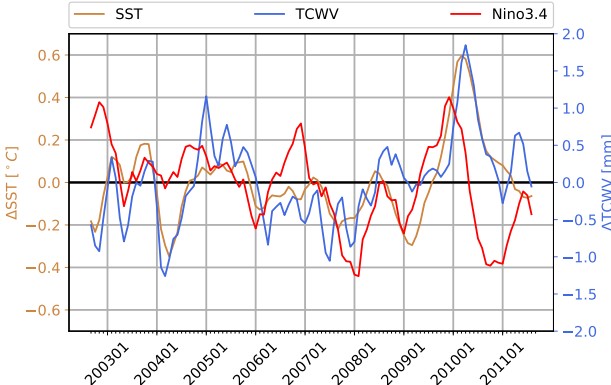

**Figure 8.** Time-series of the TAO three month running mean anomalies of SST (orange) and TCWV (blue). In red the qualitative progress of the Nino3.4-Index.

## 5 Summary and Conclusion

In this study the MODIS (aboard Aqua) cloud products were used to analyze the interannual variability of the cloud vertical distribution at the TAO for a period of 14 years (2002 to 2016). From the level 2 data, daily composites on a $0.1° \times 0.1°$ regular grid were generated. For the analysis $0.2° \times 0.2°$ monthly means, calculated on the basis of the daily composites, were used.

The data well represented large circulation patterns as the band with high cloud tops and cloud amount of the ITCZ and its seasonal shift. Also the trade wind inversion regions with its characteristic low cloud tops and the broad stratocumulus region at the west coast of Namibia were represented in the data.

The analysis of the time-series revealed strong interannual variability of the vertical cloud distribution, which was found to be mainly caused by changes in the large scale circulation due to ENSO associated teleconnective causes. The changes might

be associated with the east-ward shift and the weakening of the Walker circulation and a weakening of the Hadley cell during El Niño conditions. The largest ENSO linked anomalies were found for the HCF, which also drives the observed anomalies in the mean CTH. With an El Niño event a decrease in HCF mainly at the equatorial ITCZ influenced region was observed and was linked to a decreasing large scale vertical upward movement by including Era-Interim reanalysis data. The HCF and LCF showed opposite behaviour, which might mask much of a possible signal if using solely TCF. This supports the efforts to

consider the vertically resolved cloud fraction rather than the cloud fraction as a whole. The decrease in cloudiness is consistent with findings from e.g. Klein et al. (1999).

Furthermore, a three month lagged increase in SST and humidity during El Niño was found examining data from the AMSR-E, which is consistent with literature (e.g., Curtis and Hastenrath, 1995; Klein et al., 1999). Besides other factors as alleviated surface wind speed and a warmer troposphere the change in cloud cover is associated with positive SST anomalies e.g. Klein

et al. (1999). Subsequently, a sensitivity study of the CRF associated with ENSO linked changes could be done to quantify the



**Figure 9.** Time-series of the TAO domain-averaged HCF for different $\omega$-bins. a) $\omega$-bin1, b) $\omega$-bin2, c) $\omega$-bin3, d) $\omega$-bin4, e) $\omega$-bin5

influence of radiation changes.

As climate projection show similar changes in the tropical Atlantic circulation in a warming climate (e.g., Vecchi and Soden,





2007; Bayr et al., 2014; Hu et al., 2018), large ENSO events could be furthermore used as large "experiment" for possible effects of the global warming on the ocean-atmosphere dynamics and the CRF in the tropical Atlantic.

*Author contributions.* NM performed the scientific study and wrote the paper with support from CCH. OS processed the DARDAR data. RP and JF were involved in planning and supervised the work. All authors provided critical feedback and helped shape the research and analysis.

5 *Competing interests.* The authors declare that they have no conflict of interest.

*Acknowledgements.* This work was funded by the Federal Ministry of Education and Research in Germany (Bundesministerium für Bildungund und Forschung; BMBF) through the research programme *High Definition Clouds and Precipitation for Advancing Climate Prediction – HD(CP)$^2$*. We further thank Florian Tornow for giving valuable comments on the draft version.



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
