# Peer review of "Analysis and quantification of ENSO linked changes in the tropical Atlantic cloud vertical distribution using 14 years of MODIS observations"

_Atmospheric Chemistry and Physics, 2018_

## Referee Comment (RC1) · Anonymous Referee #1 · 12 Feb 2019

The authors use the 14-years MODIS record to explore variations in high cloud properties over the Tropical Atlantic Ocean in association with El Nino. They find trends of increasing high cloud fraction and cloud top height during the early part of the record, followed by trends of the oppose sign in the later part of the record. These changes are tied to variations in large-scale ascent in the region, which are tied to SST anomalies associated with ENSO.

I do not find the paper to be a useful contribution to the literature. It is clear from the very high correlation between high cloud fraction (HCF) and cloud top height (CTH) that both are simply measures of the relative amount of high clouds, which makes sense

intuitively but is not particularly insightful. It is well known that the amount of high clouds in a given region is strongly governed by large-scale ascent, and so tropical Atlantic HCF and CTH anomalies are unsurprisingly strongly related to the local anomalies in vertical motion. Vertical motion is this region is well known to be affected by the phase of ENSO; therefore the interannual component of temporal variations in HCF and CTH in this region will arise mostly from the phase of ENSO. Breaking the 14-year timeseries into two periods that have large ENSO-related anomalies at the endpoints will unsurprisingly introduce "trends" in vertical motion, CTH, and HCF of opposite signs in the two periods. I see no value in interpreting these as trends as opposed to variations in high cloud amount governed largely by variations in ascent.

If the authors are trying to make a more insightful point than "ENSO induces vertical motion anomalies over the Tropical Atlantic, which affects the amount of high clouds there", it is not coming through in the paper. If this is their point, then it does not rise to the level of a scientific contribution worthy of publication in this journal.

---

## Author Comment (AC1) · 21 Feb 2019

Our general response

We would like to thank reviewer 1 for reading our manuscript. We deeply regret that reviewer 1 does not consider our study insightful and the manuscript a useful contribution to the literature. Reviewer 1 does not provide specific comments on textual or technical parts in the paper. However, broader criticism on result interpretation and our linkage to large-scale/ENSO dynamics is given. The main point that we take to heart is that the scientific contribution of this study is not coming through in the paper, according to reviewer 1. Therefore, we would like to take the opportunity to argue why we consider

our study and manuscript a useful contribution to the literature by responding to the main points given by reviewer 1 as well as suggesting first adjustments, thanks to the comments of reviewer 1, to make sure that our points are insightful for everyone and, most importantly, to clarify the take-home message of the manuscript.

Our specific response

"If the authors are trying to make a more insightful point than "ENSO induces vertical motion anomalies over the Tropical Atlantic, which affects the amount of high clouds there", it is not coming through in the paper. If this is their point, then it does not rise to the level of a scientific contribution worthy of publication in this journal."

The main point of the manuscript was not to show that ENSO induces vertical motion anomalies over the TA, which affects the amount of high clouds. We are well aware that the general dynamic mechanisms of ENSO and its effect on TA ocean-atmosphere system are well described in literature. Within our work we have cited several studies discussing this. However, there are many (related) mechanisms, e.g., the response of the TA ocean-atmosphere system to a warming climate, its effect on cloud radiative forcing, convective self-aggregation, that are not well understood and quantified yet. We argue that the data and tools used within this work can contribute to this field of research. What we consider the main scientific contribution of this work is the quantification of changes in cloud properties in the TA using climatologies computed from well-established multi-annual MODIS satellite observations and at the same time being able to clearly relate the observed anomalies to (model) large-scale dynamics, and in particular, ENSO events. It would be much appreciated if reviewer 1 could provide studies/publications where satellite-observed cloud changes in the TA and their relation to large-scale dynamics have been quantified in a way that they are also useful for model studies (e.g. evaluation). If so, we would be very thankful and our study would be another contribution to this field of research, if not, we would add new measures to this field of research. In the Summary and Conclusion we state that ENSO events could serve as a large experiment for climate warming. In the future we expect a warming, possibly giving rise to similar changes in SST/water vapor/dynamics/clouds in the TA as is the case in the ENSO events in our investigated time period. We show that the satellite cloud observations of the type we used can provide an excellent tool to analyze and quantify the impact of a changing climate on clouds in the TA. To clarify our main points, we suggest several adjustments. Regarding the title, we acknowledge that it was somewhat misleading and suggest to change it from:

Observation of ENSO linked changes in the tropical Atlantic cloud vertical distribution using 14 years of MODIS observations to: Quantification of ENSO linked changes in the tropical Atlantic cloud vertical distribution using 14 years of satellite observations

Furthermore, in the Abstract, Introduction and Summary and Conclusion we will rephrase several lines to clearly emphasize that this study is about quantifying changes in satellite-observed cloud fraction and height and linking observed anomalies to large-scale dynamics/ENSO Events and not about explaining the well-known ENSO dynamics and their impact on TA.

"It is clear from the very high correlation between high cloud fraction (HC) and cloud top height (CTH) that both are simply measures of the relative amount of high clouds, which makes sense intuitively but is not particularly insightful."

Despite the fact that with MODIS it is not possible to vertically resolve clouds, using vertically resolved cloud cover, even if coarse and limited in specific situations, shows the potential of passive imagers for large-scale analysis of possible changes in the cloud vertical distribution and linking those changes to large-scale dynamics using other observations or model data. We show that the observed anomalies in the mean CTH are mainly driven by changes in the HCF and not by changes in the CTH itself, emphasizing the importance of using vertically resolved cloud cover or, in a next step, cloud types. The new insight from our study is not the change in the (high) cloud amount, but the quantification of the change in the (high) cloud amount using satellite observations. It is well-known that climate models still struggle with the correct modelling of
Interactive
comment

cloud vertical distributions, also for the TA region, giving extra importance to satellite observations, which provide cloud observations for long time periods and large spatial scales. In particular, the satellite-observed cloud anomalies and their linkage to ENSO events serve as a powerful tool to investigate and quantify the impact of a changing climate on clouds in the TA.

"I see no value in interpreting these as trends as opposed to variations in high cloud amount governed largely by variations in ascent."

We acknowledge that the use of the word trend is a bit tricky in this case. We never intended the meaning of a climatological trend. By using the two time periods and looking at temporal changes, we rather want to quantify changes due to transitions between two states. The two time periods have been chosen such that the impact of two opposite ENSO events on satellite-observed cloud properties in the TA could be quantified, mimicking possible transitions/changes due to a changing climate. In this study the use of the word trend is misleading and we will rather change to using the word transition. We will further clarify the idea behind analyzing using two time periods.

---

## Referee Comment (RC2) · Anonymous Referee #3 · 20 May 2019

The paper "Observation of ENSO linked changes in the tropical Atlantic cloud vertical distribution using 14 years of MODIS observations" deals with multiple year analysis of cloud vertical distribution parameters over Atlantic Ocean. It uses different data source, but mainly MODIS cloud products and combined radar/lidar data set DARDAR (from CPR/Cloudsat and CALIOP) to demonstrate a connection of ENSO events with cloudiness in Tropical Atlantic Ocean.

The topic meets the aim and scope of the paper.

The paper style is good, and the paper concisely written. The Figures including captions are of good quality.

[Figure]

The methods are clearly described, but some details are missing for the interpretation of the results. I recommend shortening the method sections and to remove the standard description of regression method.

The paper uses parameters describing cloud vertical distribution. In my opinion, the binned cloud fraction products (TCF, LCF, and HCF) are the much more robust parameters to show the trends and to support the main message than averaged cloud top height (CTH) due to its strongly non-Gaussian distribution.

I vote for publication after some minor comments are addressed. None of these comments are critical to acceptance.

Specific comments:

Page 1 Line 23: "Low (optically thick) .. High (optically thin)" gives the impression that low clouds are always thick and high always thin. I would add an "often" or similar. Page2, Section 2.1 MODIS: Which product do you use for the analysis and for the thresholding. The results show later cloud top height products, but the classification seems to make use of cloud top pressure. Please add this to data description for clarity. Page 3 Line 22: "much lower sampling of active lidar and radar": Calipso has footprint size of 70m and Cloudsat of 1km, so this is not much lower. Is DARDAR data set provided with a decreased resolution? Page 3 Line 31: Define TCWV and SST. Page 3 Line 30 subsection AMSR-e and Figure 8: You could have used AMSR-2 on GCOM-W for the period from 2012. This would be particularly interesting for P2 period. Figure 1: I would skip this image because you use only the two cloud pressure thresholds and not COD classifications. Section 3.2: This is for my taste too many details of standard statistical and regression methods. A reference would be sufficient. Section 3.2, page 4 Equation 1: Define variable t. Figure 2: Do you show here MODIS or DARDAR data? Section 4.1, Figure 3: Why is the MODIS and DARDAR averages that different? You discuss this: "positive bias towards higher CF's which can be explained with the higher sensibility of lider for high, optically thin clouds.": But the difference is also big for low

clouds. (0.2 vs 0.45)! Page 7, line 9: lider should be lidar Page 7 Line 6: "The LCF and CTH show a negative correlation. This indicates that the mean seasonal variability of the CTH is mainly related to changes in the vertical cloud distribution.": I think this is a tautology. The cloud top height is directly taken from the vertical cloud distribution. Figure 7: I cannot see plus signs in the images in my print-out and hardly in the pdf. And they look very similar to the grid lines.

---

## Author Response (AR1)

RESPONSE to comments from reviewer #3:

We would like to thank reviewer #3 for reading our manuscript and giving specific comments on textual or technical parts in the paper.

**I recommend shortening the method sections and to remove the standard description of regression method.**
We agree with the reviewer that the method section can be shortened.
Following equations have been removed: eq. 7, 8, 9, 11

**The paper uses parameters describing cloud vertical distribution. In my opinion, the binned cloud fraction products (TCF, LCF, and HCF) are the much more robust parameters to show the trends and to support the main message than averaged cloud top height (CTH) due to its strongly non-Gaussian distribution.**
We agree with your comment. Although we mention several times in the manuscript that changes in the averaged CTH are not due to real changes of the cloud top, but are mainly due to more or less high clouds. We propose to emphasize the non-gaussian character of the CTH again.
Text has been adopted accordingly on p. 6, l.23

**Page 1 Line 23: "Low (optically thick) .. High (optically thin)" gives the impression that low clouds are always thick and high always thin. I would add an "often" or similar.**
For clarity this has been adapted (p.2, l. 2-3).

**Section 2.1 MODIS: Which product do you use for the analysis and for the thresholding. The results show later cloud top height products, but the classification seems to make use of cloud top pressure. Please add this to data description for clarity.**
We used the MODIS pixel based cloud height categories flag for the calculation of the vertically resolved cloud fraction. This is based on the CTP and consistent with the ISCCP classification scheme. For our analysis we used the CTH and the computed vertically resolved cloud fraction. This was clarified in the data description section on p.4, l.24.

**Page 3 Line 22: "much lower sampling of active lidar and radar": Calipso has footprint size of 70m and Cloudsat of 1km, so this is not much lower. Is DARDAR data set provided with a decreased resolution?**
Calipso and Cloudsat has only nadir view, leading to much lower daily spatial coverage.

**Page 3 Line 31: Define TCWV and SST.**
Defined on p.2, l.26-27.

**Page 3 Line 30 subsection AMSR-e and Figure 8: You could have used AMSR-2 on GCOM-W for the period from 2012. This would be particularly interesting for P2 period.**
Thank you for pointing this out. We will consider this in a follow-up study where the time series will be extended.

**Figure 1: I would skip this image because you use only the two cloud pressure thresholds and not COD classifications.**
We agree, the image has been removed and the classification is described in the text, see lines p.4, l. 22-26.

**Section 3.2: This is for my taste too many details of standard statistical and regression methods. A reference would be sufficient.**
We agree, several equations have been removed, references to the methods are given.

Changes were made in p.4 and 5.

**Section 3.2, page 4 Equation 1: Define variable t.**
Variable t is now defined (p.5,l .1)

**Figure 2: Do you show here MODIS or DARDAR data?**
We show MODIS data in this figure. It has now been clarified in the figure caption.

**Section 4.1, Figure 3: Why is the MODIS and DARDAR averages that different? You discuss this: "positive bias towards higher CF's which can be explained with the higher sensibility of lider for high, optically thin clouds.": But the difference is also big for low clouds. (0.2 vs 0.45)!**
We agree with your comment. As DARDAR can have both low and high clouds in the same column, the differences can be explained by multi-layer conditions (high + low) that can be up to 25 to 30% of the cloud cover, especially in the tropics.
Has been adapted on p.7, l. 3-4.

**Page 7, line 9: lider should be lidar Page 7**
Has been corrected.

**Line 6: "The LCF and CTH show a negative correlation. This indicates that the mean seasonal variability of the CTH is mainly related to changes in the vertical cloud distribution.": I think this is a tautology. The cloud top height is directly taken from the vertical cloud distribution.**
We agree with your comment. Phrase has been taken out.

**Figure 7: I cannot see plus signs in the images in my print-out and hardly in the pdf. And they look very similar to the grid lines.**
Has been changed, now dost are used.

Second RESPONSE to comments from reviewer #1:

Regarding the comments of reviewer #1, we changed the title from „*Observation of ENSO linked changes in the tropical Atlantic cloud vertical distribution using 14 years of MODIS observations*" to „*Analysis and quantification of ENSO linked changes in the tropical Atlantic cloud vertical distribution using 14 years of MODIS observations*".

Furthermore we made several changes in the Abstract, Introduction and Summary to clearly emphasize that the study is about quantifying changes in the passive satellite-observed vertical cloud fraction and height and linking them to larg-scale dynamics/ENSO events.

Moreover we changed the phrase in p.9 l.9 from *"For the analyzed time period no consistent significant linear trend was found for the domain average (not shown)."* to *"For the analyzed time period significant (p-value < 0.05) linear changes were found for the domain-averaged CTH (178 m/dec), HCF (0.0006 /dec) and the LCF (0.001 /dec)."*

The caption of Figure 5 has been changed form „*Time series of the tropical Atlantic means...*" to „*Quantification of linear changes in the tropical Atlantic mean associated with ENSO phase transition.*"

On p.12 l.6 we changed the phrase „*As shown in (Sec. 4.2 & 4.3) a link between ENSO and TOA 
[revised manuscript text omitted]

---

## Author Response (AR2)

**RESPONSE to comments from reviewer #1:**

*We would like to thank reviewer #1 for the revision and the critical comments and feedback to our manuscript.*

**1) The first paragraph of the introduction confuses cloud radiative forcing (or cloud radiative effect) with the cloud radiative feedback. Cloud radiative effect quantifies the radiative effect of clouds on the mean-state of the climate; the cloud feedback quantifies how this radiative effect changes per unit global temperature change. Hence the first sentence of the introduction is incorrect, as is the sentence starting on Line 3 of Page 2.**

*1)* Your right. *That's an important point and has been adapted.*

**2) I don't understand the analysis described on page 9: "As we look here at effects associated with ENSO, we used the MLRM without Nino3.4 index (see Sec. 3.2)." What does "here" mean? It is never explicitly stated how the Nino3.4 index is included in the MLRM in the first place, and it is unclear why one would remove it from the MLRM in order to look at its effects.**

*We see your point and it's misleading.*

*We also analyzed the time-series over the whole time-period where we included the Nino3.4 index to see possible changes associated with a warming climate. But this was not the focus of the study and is only shortly described on page 9, line 6-13. With "here" we wanted to say, that in the main analysis, where we analyzed the two periods associated with the transition between the two ENSO states, we didn't include the Nino3.4 index because we wanted to explicitly consider the influence of the variability of the ENSO on the variables and their linear change.*

*We rephrased this part to:*

*"For this analysis, we used the MLRM explained in Sec. 3.2".*

**Further, I don't understand what you are doing in this statement: "To find the best separation of the phases the maximum of the sum of the explained variances of the two models was calculated for differing phase separations."**

*To decide which is the most suitable separation of the time-period, we used the month where both models for phase 1 and phase 2 fits best. This means, that we used the month for the separation where the sum of the explained variances of the two models has its maximum.*

**3) There are many grammatical errors throughout, including:**
**a. P3,line26: should be "additional"**          *adjusted*
**b. P4, line 22: subject-verb agreement**         *adjusted*
**c. P6, lines11-12: phrasing is odd "roughly….HCF"** *roughly → approximately*
**d. P7: unclear what "major phases" means**     *adjusted*
**e. P8, line 2: "vise verse"**               *adjusted*
**f. P8, line 14: should be "opposing"**         *adjusted*
**g. P9, line 2: should be "where"**            *adjusted*
**h. P10, line 10: "is despite of the low"**      *adjusted*
**i. Table 1: "ascend" should be "ascent"**      *adjusted*

*The mentioned grammatical errors have been adapted.*

[revised manuscript text omitted]